# Maternal Exposure to Indoor Air Pollution and Birth Outcomes

**DOI:** 10.3390/ijerph16081364

**Published:** 2019-04-16

**Authors:** Peter Franklin, Mark Tan, Naomi Hemy, Graham L. Hall

**Affiliations:** 1School of Population and Global Health, Faculty of Health and Medicine Sciences, The University of Western Australia, Crawley, WA 6009, Australia; 2School of Paediatrics and Child Health, Faculty of Health and Medicine Sciences, The University of Western Australia, Crawley, WA 6009, Australia; mark@solutions4.me; 3Telethon Kids Institute, Nedlands, WA 6009, Australia; Naomi.Hemy@telethonkids.org.au (N.H.); Graham.Hall@telethonkids.org.au (G.L.H.); 4School of Physiotherapy and Exercise Science, Curtin University, Bentley, WA 6102, Australia

**Keywords:** indoor air pollution, formaldehyde, pregnancy, birth outcomes

## Abstract

There is a growing body of research on the association between ambient air pollution and adverse birth outcomes. However, people in high income countries spend most of their time indoors. Pregnant women spend much of that time at home. The aim of this study was to investigate if indoor air pollutants were associated with poor birth outcomes. Pregnant women were recruited prior to 18 weeks gestation. They completed a housing questionnaire and household chemical use survey. Indoor pollutants, formaldehyde (HCHO), nitrogen dioxide (NO_2_) and volatile organic compounds (VOCs), were monitored in the women’s homes at 34 weeks gestation. Gestational age (GA), birth weight (BW) and length (BL) and head circumference (HC) were collected from birth records. The associations between measured pollutants, and pollution surrogates, were analysed using general linear models, controlling for maternal age, parity, maternal health, and season of birth. Only HCHO was associated with any of the birth outcomes. There was a 0.044 decrease in BW z-score (*p* = 0.033) and 0.05 decrease in HC z-score (*p* = 0.06) for each unit increase in HCHO. Although HCHO concentrations were very low, this finding is consistent with other studies of formaldehyde and poor birth outcomes.

## 1. Introduction

Exposure to toxicants during pregnancy has the potential to affect foetal growth and development [1]. There is increasing evidence to suggest that prenatal exposure to ambient air pollution is associated with adverse birth outcomes such as preterm delivery, low birth weight (LBW) and small for gestational age [2,3,4,5,6,7,8,9,10], although the findings are not consistent [11,12].

Much of the research has examined the implications of exposure to outdoor air pollution. There has been limited research into the impact of indoor air pollution (IAP). Studies on IAP have tended to focus on the detrimental impacts of the burning of biomass for cooking and heating in low to middle income countries [13]. However, IAP is also a cause for concern in high income countries (HIC), albeit with less severe consequences, as people spend most of their time indoors and the indoor environment can be a major source of personal exposure to many pollutants [14].

For pregnant women the majority of time is spent indoors at home [15,16,17,18] and this increases as pregnancy progresses [17,18]. Despite this there has been very little research into maternal exposure to IAP in HIC and birth outcomes [19]. The aim of this study was to explore the association between indoor air exposures during pregnancy and birth outcomes, specifically gestational age, birth length, weight and head circumference.

## 2. Methods

### 2.1. Subjects and Protocol

Non-smoking pregnant women were recruited from within a larger birth cohort study of the effect of maternal stress responses to physical, biological and emotional stressors on foetal and child health outcomes. Women were recruited prior to 18 weeks of gestation. At initial recruitment, non-smokers were approached and asked if they wished to participate in the IAP component. Those that agreed completed a number of validated questionnaires, including a housing inventory and chemical usage survey, at recruitment. Indoor air concentrations of formaldehyde (HCHO), nitrogen dioxide (NO_2_) and volatile organic compounds (VOCs) were monitored at 34 weeks gestation. Data on traffic near the women’s home address as well as exposure to environmental tobacco smoke (ETS) was also collected. Standard demographic variables, including gestational age (GA), length (crown-heel length, CHL), weight (BW) and head circumference (HC) were obtained from birth records. Ethics approval for this was obtained from both the Princess Margaret Hospital (Registration number 1611/EP) and Murdoch University (Project number 2007/238) human research ethics committees.

### 2.2. Techniques

#### 2.2.1. Housing Inventory

The housing inventory included questions relating to house type, heating and cooking systems, and renovations in the past 12 months. This was modified from the housing inventory published by Lebowitz et al. [20]. The women also completed a survey of their use of household chemical products within the past 12 months, which was adapted from the Avon Longitudinal Study of Parents and Children (ALSPAC) [21]. Household chemicals included; disinfectant, bleach, carpet cleaner, window cleaner, dry cleaning fluid, aerosols, turpentines/white spirits, air fresheners, paint stripper, paint/varnish and pesticides. Frequencies for usage of each product were recorded as being rarely, monthly, fortnightly, weekly, most days or daily [21].

#### 2.2.2. Indoor Air Monitoring

Each of the indoor pollutants, HCHO, NO_2_ and VOCs, were monitored for 7-days using validated passive sampling techniques. Formaldehyde was monitored in both the bedroom and main living room using 37 mm passive sampling badges containing glass fibre filters impregnated with 2,4-dinitrophenylhydrazone (DNPH) solution [22]. Formaldehyde was analysed using High Performance Liquid Chromatography (HPLC, Varian Prostar Model 430, Varian Inc., Palo Alto, CA, USA). Nitrogen dioxide sampling was based on a method described by Nitschke et al. [23]. Sampling was conducted in the kitchen using 50 mm diameter passive sampling badges containing 47 mm cellulose filters impregnated with triethanolamine (TEA) solution. Analyses were done using spectrophotometry (UV 1601, Shimadzu, Tokyo, Japan). Volatile organic compounds were measured in the main living room of the home using stainless steel passive sampling tubes with a Tenax sorbent (ATD Tenax TA 60/80, Perkin-Elmer, Boston, MA, USA). The tubes were analysed at a nationally accredited chemistry laboratory using thermal desorption gas chromatography and mass spectroscopy (GC/MS) according to USEPA method TO-17 [24]. For all pollutants, half the limit of detection (LOD) was used when the values were below LOD. The LOD for each pollutant were: 1.9 ppb (2.4 µg/m^3^) for formaldehyde; 0.42 ppb (0.8 µg/m^3^), and 0.5 ppb for individual VOCs

#### 2.2.3. Other Data

Data was collected for exposure to ETS and traffic, as well as maternal health status. Exposure to ETS was determined by questions on whether there were smokers in the home and if the mothers were exposed to 2 or more hours of ETS at home and/or at work. Distance from home to main roads was used as a proxy for traffic pollutant exposure. This was obtained by geocoding each woman’s home address and mapping these on road maps obtained from Western Australia Main Roads Department using a specialised computer software package (ARC GIS version 10, ESRI, Redlands, CA, USA, 2010). Details of current and past health of the women were collected from questionnaires filled in by the mothers at 18 and 34 weeks of gestation. Data was collected on current asthma, current and gestational diabetes, and current and gestational high blood pressure. These were all self-reported. Maternal age at birth of the child, parity and season of birth were also recorded.

### 2.3. Statistical Analysis

#### 2.3.1. Treatment of Data

*Birth outcomes*: Gestational age was normally distributed so the raw data was used in the analyses. All other birth measures, BW, BL and HC, were transformed to z-scores based on birth centile charts from the WHO [25]. The z-scores take into account sex and gestational age of the newborn

*Exposure data*: Pollution concentrations could not be transformed to normal so raw values were used. For HCHO, the average of bedroom and living room concentrations was used. For VOCs, a suite of 31 compounds were monitored but most were not detected. The most commonly detected compounds were the BTEX compounds (benzene, toluene, ethyl-benzene and xylenes) so a BTEX score was calculated by summing up those compounds. A total household chemical use score (composite household chemical exposure (CHCE) score) was calculated. This score was calculated as a sum of reported usage frequencies of each category of the 11 household products. The scoring guide to each product based on frequencies of used were as follows: rarely (0); monthly (1); fortnightly (2); weekly (3); most days (4) or daily (5) [21]. A score between 0 and 55 was possible. CHCE scores were normally distributed. Finally, distance from main roads was categorised into three categories, <50 m, 50–100 m, and >100 m. Exposure to ETS was treated as a binary variable (<2 h/day and ≥2 h per day)

*Other variables*: Maternal age was normally distributed. Maternal health conditions (asthma, diabetes, epilepsy, elevated blood pressure) were classified as binary variables (present/not present). For both diabetes and hypertension, current and gestational conditions were combined for a single variable. There were three categories for parity: nulliparous, primipara, and multipara.

#### 2.3.2. Data Analyses

The relationship between IAP exposures and each of the four birth outcomes (GA and BW, BL and HC z scores) were analysed using general linear models. Separate models were constructed for each indoor air exposure variable, as well as distance from main road and exposure to ETS. All models included maternal age and parity as both of these factors are known to affect birth outcomes [26]. Maternal asthma [27], diabetes [28] and blood pressure [29] can also affect birth outcomes, as can season of birth [30], and these factors were also included *de novo* in models. Distance from main road and exposure to ETS were included in each of the models of measured pollutants but removed if they were not significant at the level of 0.5. All analyses were conducted using SPSS (IBM SPSS Statistics V25, IBM Corp., Armonk, NY, USA)

## 3. Results

### 3.1. Participants

Three hundred and seventy-three non-smoking women (373) were approached and 305 (81.7%) agreed to participate in the IAP sub-study. Of these participants, birth data were available for 266 births. Furthermore, four women were found to be active smokers and were excluded from the analyses. Therefore, the final analyses included 262 live births (50.8% female). Most births were full term with only 6 babies born <37 weeks, but all born after 36 weeks. The age range for mothers was 18 to 41 years (mean 29.6 years) and most were healthy with very few reported health issues (Table 1). Details for maternal age, parity, and maternal health, as well as gestational age, birth length, birth weight and head circumference are presented in Table 1. Parity was associated with gestational age, with GA decreasing with increasing parity. No other maternal variable (age or health status), nor season of birth, were associated with birth measures.

### 3.2. Environmental Pollutants

Indoor air pollution concentrations were low with a large number of samples below the LOD, particularly for NO_2_ and VOCs. There were no significant correlations between measured IAPs (data not shown). Fifty-eight women reported significant (>2 h) daily exposure to ETS. Only 9 homes were within 50 m of a major road, while a further 14 were within 100 m. Indoor air data, as well as the distance of homes (by categories) from main roads, are presented in Table 2.

### 3.3. Associations between Environmental Pollutants on Birth Outcomes

*Gestational age*: None of the environmental pollutants, monitored and surrogates, were associated with GA.

*Birth weight*: Average household HCHO was associated with reduced BW. After controlling for parity, maternal age and maternal health, there was a decrease of 0.044 z-score in birth weight for every 1 µg/m^3^ increase in HCHO (*p* = 0.033) (Table 3). No other exposure variable was associated with birth weight (Table 3).

*Birth length*: There was no association between BL and any of the exposure variables.

*Head circumference*: Of all the exposure variables, only HCHO was associated with HC. There was a decrease of 0.056 z-scores for every 1 µg/m^3^ increase in average household HCHO, but this was not statistically significant (*p* = 0.06).

## 4. Discussion

There are only a few studies on the associations between maternal exposure to IAPs on birth outcomes. In this study there was a significant association between HCHO concentrations in homes and reduced BW. Neither of the other measured pollutants, NO_2_ and VOCs, nor the pollution surrogates, CHCE, distance from main roads and ETS, were associated with any birth outcomes. Although HCHO concentrations in homes were low, the results are consistent with other data on HCHO and birth outcomes.

Although pregnant women in high-income countries spend most of their time indoors, particularly toward the end of pregnancy [17,18], there are very few studies investigating whether indoor exposures affect pregnancy and birth outcomes [19]. A few studies have assessed personal exposure of pregnant women to various pollutants, including polycyclic aromatic hydrocarbons (PAH) [30], benzene [31], chlorpyrifos pesticides [9] and fine particulate matter (PM2.5) [7], over a 2–7 days period. Personal exposures capture indoor as well as outdoor concentrations and cannot specifically explore associations between indoor air pollutants and birth outcomes. Associations between these pollutants and reduced BW, BL and HC have been observed in these studies [7,9,31,32]. Although total exposure is likely to be more important than ‘compartmentalised’ exposures, it is not known what the main source of exposures to these pollutants were, although for PAHs, benzene and PM2.5, outdoor sources, particularly traffic, are probably the most important.

Of the pollutants measured only HCHO was associated with any of the birth outcomes. Although HCHO is a constituent of outdoor air it is generally found in higher concentrations indoors [14] and indoor levels are strongly associated with personal exposures in both adults [33] and children [34]. Formaldehyde has been associated with adverse pregnancy and birth outcomes such as reduced fertility, spontaneous abortion, foetal growth, birth malformations and low birth weight [35,36]. Most of the evidence comes from occupational exposures and animal studies [36,37], although a few studies have observed adverse effects of non-occupational HCHO exposures on pregnant women [35,38]. Maroziene and Grazuleviciene [38] observed reduced birth weight associated with maternal exposure to increased concentrations of formaldehyde in outdoor air, while Amiri and Turner-Hensen [35] found that increased personal exposure (combined indoor and outdoor exposure) of pregnant women to HCHO during the second trimester of pregnancy was associated with decreased biparietal diameter, measured by ultrasound, at that time.

The concentrations of HCHO in the current study are low (mean 2.81 µg/m^3^), well below occupational exposures and residential concentrations that have previously been associated with, mostly respiratory, health effects [39,40,41]. However, the data are consistent with the study of Maroziene and Grazuleviciene [38], which found an association between low outdoor HCHO concentrations (mean 3.14 µg/m^3^) and reduced BW [38].

The mechanism(s) that may explain the effects of HCHO on the developing foetus are not known but may include genotoxicity, oxidative stress, disruption of the protein, enzyme and hormonal activity and DNA methylation [37]. Formaldehyde can cross the placenta and enter foetal tissues [42]. Compared to maternal tissue, foetal tissue can more readily take up HCHO, as well as eliminate it much more slowly [43]. There is evidence that HCHO is associated with oxidative stress in animal foetal cells and tissues [44], which may explain the relationship between formaldehyde exposure and foetal growth [35]. Whether chronic low-level exposure to HCHO in homes is sufficient to elicit these responses requires further investigation.

There were no associations with other pollutants, measured directly or as surrogates, and birth outcomes. The lack of associations in this study may be due either to the low levels of those pollutants, limitations of the exposure assessment, or an association does not exist. Indoor concentrations of measured pollutants were low, with many samples for NO_2_ (46.6%) and VOCs (19.4%) below the limit of detection. Exposure to ETS was self-reported and there was no objective measure of exposure, such as cotinine, obtained in this study. Outdoor air pollution was assessed by proxy using distance from roads. The study was conducted in a small city, and its surrounding rural areas, with a population of about 80,000. There is no major industry in the area and the main contributor to outdoor air pollution is traffic. However, traffic is only relatively heavy during the holiday season and very few homes of the pregnant women were within 100 m of roads that could get intermittent heavy traffic flows.

Both NO_2_ [6,45,46,47,48] and benzene, or BTEX, [2,44] as outdoor air pollutants, have been associated with reduced BW and other birth outcomes. In the current study these pollutants were only measured indoors, which may not adequately reflect personal exposure to these pollutants. Furthermore, as these pollutants are only measured at a single time point they may not reflect typical exposure levels across pregnancy. Of the pollutants measured, HCHO has the least temporal variation. A single 3-day measure has been reported to represent the mean monthly HCHO concentration [49]. Gavin et al. [49] recommended at least a summer and winter measure would be a reasonable approach to estimate annual mean HCHO and a number of studies in Australian homes have reported seasonal variation [41,50,51,52]. However, these variations are small. Furthermore, HCHO levels vary little between rooms inside homes [49,50]. The other pollutants, NO_2_ and VOCs, however, may vary considerable both spatially and temporally, and are dependent on sources. For example, NO_2_ concentrations can increase rapidly with the use of gas appliances [53] and there can be large inter-room variations within homes [54]. Indoor concentrations of individual VOCs are also likely to be influenced strongly by specific sources. For example, indoor BTEX concentrations predominantly reflect outdoor air and traffic and varies diurnally depending on traffic [55].

There are a number of limitations to this study. The sample size was small (*n* = 266), although this is a similar size to other pregnancy exposure studies, such as studies conducted in the USA (*n* = 263) [9], Poland (*n* = 362) [7] and France (*n* = 273) [31]. Monitoring was conducted indoors only and there was no personal monitoring. We did not have data on where the individual participants spent most of their time during the monitoring period. However, previous research has suggested that indoor residential concentrations of HCHO are a good predictor of personal exposures [32,33] and pregnant women spend an increasing time indoor at home as pregnancy progresses [17,18]. Data was not available for all potential risk factors or confounders, such as diet, other environmental exposures, maternal BMI and maternal health, that could influence the results. The monitoring was limited to the third trimester so there was no way of determining trimester effects or whether this is the most important period for exposure to the selected pollutants. The study by Maroziene and Grazuleviciene [37] found only first trimester exposure to ambient formaldehyde was associated with lower birth weight. However, data on trimester effects have varied considerably and different contaminants can effect foetal development at different time periods [11]. Trimester effects are an important issue that need to be considered in future studies of IAP. Finally, there were a large number of analyses conducted so the chance of a type II error was high. The findings for HCHO in this study may well be due to chance alone. However, there are biological plausible mechanisms for the findings, although whether these are relevant at such low exposure levels is uncertain.

## 5. Conclusions

This study provides evidence that indoor exposures, even at low levels, may be associated with adverse birth outcomes. Formaldehyde, which is predominantly an indoor air pollutant, was associated with reduced birth weight. These findings are consistent with other research, albeit mostly involving much higher exposures. Further research is required to attain a better understanding of how low-levels of HCHO exposure may influence foetal development.

## Figures and Tables

**Table 1 ijerph-16-01364-t001:** Details of mothers and newborns.

**Mother**	
Maternal age (years) *	29.6 (5.3)
Parity	
nulli	32.6%
primi	40.7%
multi	26.7%
Maternal health ^#^	
Asthma	4.1%
Diabetes	6.9%
High blood pressure	8.4%
**Child**	
Sex (male/female)	129/133
Gestational Age * (weeks)	38.97 (1.04)
Weight * (kg)	3.52 (0.43)
z-score	0.44 (0.88)
Length * (cm)	51.35 (2.51)
z-score	0.97 (1.31)
Head circumference * (cm)	34.89 (1.46)
z-score	0.57 (1.16)

* Mean (SD); ^#^ %yes.

**Table 2 ijerph-16-01364-t002:** Concentrations and proportions of measured pollutants and pollution surrogates.

Pollutant/Mixture		Proportion < LOD (%)
**1. Monitored pollutants**		
Formaldehyde (µg/m^3^) *	2.81 (LOD–17.33)	23.3
Nitrogen Dioxide(µg/m^3^) *	0.76 (LOD–70.57)	46.6
BTEX (ppb) *	3.00 (LOD–145.00)	19.4
**2. Pollution surrogates**		
CHCE ^#^	16.85 (0–33)	
ETS (% exposed)	22.8%	
Distance to road		
<50 m	3.5%	
51–100 m	5.4%	
>100 m	91.2%	

* Median (range); ^#^ Mean (range); LOD—Limit of Detection; CHCE—Combined Household Chemical Exposure; ETS—Environmental Tobacco Smoke.

**Table 3 ijerph-16-01364-t003:** Associations between environmental exposures and birth weight.

Variable	β-Coefficients (95% CI)	
	Unadjusted	Adjusted *
HCHO ^†^	−0.032 (−0.067, −0.008)	−0.044 (−0.085, −0.004)
NO_2_ ^†^	−0.003 (−0.014, 0.007)	0.002 (−0.010, 0.014)
BTEX ^†^	−0.003 (−0.011, 0.005)	−0.004 (−0.014, 0.005)
CHCE ^†^	−0.008 (−0.024, 0.009)	−0.009 (−0.025, 0.008)
ETS		
Yes	−0.097 (−0.355, 0.162)	0.164 (−0.108, 0.435)
No	ref	ref
Distance from roads		
<50 m	−0.348 (−0.937, 0.242)	−0.387 (−1.020, 0.245)
51–100 m	0.062 (−0.416, 0.539)	−0.011 (−0.523, 0.501)
>100 m	ref	ref

* Adjusted for maternal age, parity, maternal asthma, maternal diabetes, maternal hypertension and season of birth; ^†^ z-score change per unit increase in pollutant.

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
