# Peer review of "Maternal Exposure to Indoor Air Pollution and Birth Outcomes"

_ijerph, 2019, doi:10.3390/ijerph16081364_

Round 1

Reviewer 1 Report

The IJERPH Number of this manuscript is ijerph-423248-peer-review-v1.

The authors have revised the manuscript according to the reviewer’s suggestion.  What I concern have been added in the text, including the description of study subjects and the indoor air monitoring methodology. Therefore, I suggest it can be accepted.

Reviewer 2 Report

The authors have addressed my concerns and I do not have further comments.

Reviewer 3 Report

Yes, the authors have addressed these comments.

This manuscript is a resubmission of an earlier submission. The following is a list of the peer review reports and author responses from that submission.

Round 1

Reviewer 1 Report

This study assessed the associations between indoor air pollutant and birth outcomes, and found formaldehyde concentration in indoor air was associated with poor birth outcomes. Below are some comments.

1. Line 57 The housing inventory included questions relating to house type, distance from main roadways. However, in line 84, it was obtained by geocoding each woman’s home address and mapping these on road maps. Please clarify it.

2. Line 112-119 Data analyses, The authors included confounders all based on previous studies. Were some covariates included due to the association with exposures or outcomes in their own analyses?

3. Line 115-116 Maternal asthma, diabetes and blood pressure. Were they diabetes or hypertension before pregnancy or gestational diabetes/hypertension? What is the cutoff value of hypertension in this study?

4. Table 1, maybe the characteristics left align will be easier to read.

5. Line 135 Indoor air pollution concentrations were mostly low with a large number of samples below the LOD. The authors should present the LOD of each pollutant. Was the LOD of each pollutant comparable with LOD in other studies?

6. Line 142, In this epidemiological study, “the associations” will be better than “the effect”

7. Line 147-150 These results could not be found in Table 3. Also, ”0.047 and 0.37 for non-exposed and exposed”, were 0.047 and 0.37 were beta value? Then how did the authors know birth weight was lower for children of women exposed to ETS?

8. The authors have the data on Frequencies for usage of household chemicals and the chemical concentrations. They are advised to analyze the relationships between them.

9. Line 160 “There are very few studies on the effects of maternal exposure to IAPs on birth outcomes.” Actually, there are some such studies, not very few.

10. Line 176 Although HCHO is a constituent of outdoor air. It is not right. HCHO is an indoor air pollutant.

11. It seems that the how long the pregnant women stay in home was not included in the analyses. 

Reviewer 2 Report

The manuscript presents the results of association between birth outcomes and maternal exposure to indoor air pollution at relatively lower level.  It is useful data for the risk assessment on indoor pollution. However, the following points should be noted:

How can we understand “Non-smoking pregnant women were recruited from within a larger birth cohort study” (line 45)?  Your study subjects come from this larger birth cohort?   If so, how many subjects have been recruited totally? 

When you do the indoor air monitoring, how many samples have you taken in each family for specific pollutant?  If not 24ours, how long is the sampling time? And at what time period (e.g. 8:00am to 8:00pm) is it sampled?

Please give more information on exposure level, while you discussed the association between formaldehyde exposure and birth outcomes (line 178 to 186). 

Reviewer 3 Report

The manuscript (MS) “Maternal exposure to indoor air pollution and birth outcomes” is a study to investigate that the relationship between indoor air pollutants and poor birth outcomes. This study, found that HCHO was associated with decrease BW z-score and HC z-score. But the paper has some limitations:

Some specific comments are as followed:

Introduction: Are there any studies focus on both indoor and outdoor pollution? What's the relationship between them?  It is appropriate to summarize the results of other researches in animal models or population studies. These need to be clearly described in the background.

Statistical analysis: Have you calculated the minimum sample size that reflects the model's power, and whether 262 pairs of maternal and infant pairs have statistical power. Why not try to transform Pollution concentrations to normal distributed data?

Result: When you analyzed the associations between environmental exposures and birth weight, you should show the crude model and the adjusted model, and why not adjusted gestational age?

Discussion: Your  data are consistent with the study of Maroziene and Grazuleviciene, which found an association between low outdoor HCHO concentrations and reduced BW. What's the difference between your research and his? What is your innovation?

Discussion: The measure used in the paper was questionable. What about mother's exposure to pollution at other places, such as working places, other than her residential address? Did the authors assume that the mother stayed at home during the entire pregnancy? This should be discussed.

Table 3. Associations between environmental exposures and birth weight. I cannot see the results about ETS and the distance between women and major road in Table 3, and there should be an abbreviation in the note.

Reviewer 4 Report

This study was conducted to investigate the impacts of maternal exposure to indoor air pollution on birth outcomes. A total of 262 pregnant women were analyzed in the study. Maternal exposure to indoor air pollutants were assessed at 34-week gestation. The study found that maternal exposure to formaldehyde was negatively associated with birth weight and head circumference. A few suggestions and concerns are discussed below.

A specific birth outcome may mainly be affected at a window of time during pregnancy. As indoor air pollutants were only monitored for either 24-hour or one-week period using passive sampling techniques at 34-week gestation, the selected periods for IAP monitoring may not be the right windows of exposure for some birth outcomes.  Please discuss if some non-significant observations are due to this limitation.

The study was conducted to study the effects of indoor air pollution exposure. However, as there was no monitored data on ambient air pollution in the study, it is unable to control for the effects of ambient air pollution exposure when the authors investigated the effects of IAP.

In the final regression models, it seems only controlling for maternal characteristics. Please clarify the strategies which were used to select the variables in the final regression models in the section of statistical analysis.

In this study, several air pollutants have been monitored. It may be interesting to assess the combined effects of multiple indoor air pollutants on adverse birth outcomes.

Reviewer 5 Report

There are too many limitations of the exposure assessment to draw meaningful conclusions about the indoor air pollution exposure and birth outcomes. In fact, authors indicate their major finding regarding the association of formaldehyde and birth weight could be due to chance. Major issues with this study are detailed below: 

1.    There is not enough information about the participant recruitment. The authors mention this is part of larger birth cohort study, but do not give details or an additional reference to this study. Readers need to know details such as inclusion/exclusion criteria, time frame of recruitment, and geographic location of participants recruited. Overall, there is insufficient information about participant selection.    

2.    The exposure assessment methods to determine indoor air concentrations of pollutants were not robust. Mainly, the low temporal coverage, only 24 hours for NO2 and 7 days for HCHO and VOCs, is inadequate to characterize exposure across trimester, much less pregnancy. Moreover, the spatial coverage is also inadequate, since pollutant concentrations are measured disparately across the home environment (i.e., kitchen versus living versus living and bedroom). There is no information on the occupation of these participants and mobility. The effects of individual mobility would significantly influence the individual-level exposure. 

3.    Potential confounding was not adequately addressed. While maternal age, parity, maternal health, and season of birth was adjusted for in models, other important factors were not considered. Namely, social class, which is strongly associated with birth outcomes and pollutant exposure was not addressed. There is no information on maternal race/ethnicity, although we can assume they are Australian, it is unclear to readers. ETS was measured through a questionnaire, and reported that 58 women were exposed to >2h ETS. While there was a non-significant association between ETS exposure and birthweight, this was not adjusted for in the model examining pollutant exposure and birth outcomes.